# Reverse T$_3$ in patients with hypothyroidism on different thyroid hormone replacement

**Julian B. Wilson**[1], **Thanh D. Hoang**[2], **Martin L. Lee**[1,3], **Ma'ayan Epstein**[1,3], **Theodore C. Friedman**[1,4]*

1 Division of Endocrinology, Metabolism and Molecular Medicine, Department of Internal Medicine, Charles R. Drew University of Medicine and Science, Los Angeles, California, United States of America, 2 Endocrinology Division, Walter Reed National Military Medical Center, Bethesda, Maryland, United States of America, 3 UCLA Fielding School of Public Health, Los Angeles, California, United States of America, 4 David Geffen School of Medicine at University of California, Los Angeles, California, United States of America

* theodorefriedman@cdrewu.edu

## Abstract

### Background

Reverse T$_3$ (rT$_3$) is a biologically inactive form of T$_3$ (triiodothyronine), a thyroid hormone, that is created by peripheral 5 deiodination of T$_4$ (thyroxine) by type 1 and type 3 deiodinase enzymes (D1 and D3 respectively) and may block T$_3$ binding to the thyroid hormone receptor. Approximately 15% of patients on L-T$_4$ replacement therapy with a normalized thyroid-stimulating hormone (TSH) report experience continued fatigue and other hypothyroid symptoms; therefore, efforts are needed to understand why this occurs and how it can be corrected. Decades ago, endocrinologists realized that in patients with severe illnesses, rT$_3$ is typically high and T$_3$ is typically low; this was termed "euthyroid sick syndrome". More recently, functional medicine and other doctors, have argued that high rT$_3$ is detrimental and can block T$_3$ from binding to the thyroid hormone receptor. Due to the lack of peer-reviewed publications on this topic, functional medicine doctors continue to rely heavily on rT$_3$ levels to treat patients that may have no other laboratory findings of hypothyroidism and often prescribe L-T$_3$-only preparations to patients in an effort to lower rT$_3$.

### Methods

The initial rT$_3$ measurements done by liquid chromatography/tandem mass spectrometry (LC/MS-MS) were retrospectively analyzed from the initial blood tests in 976 consecutive patients, with symptoms of fatigue and treated for hypothyroidism, in a private Endocrinology practice. TSH, free T$_3$ and free T$_4$ were measured by electrochemiluminescence immunoassay (ECLIA). The upper limit of normal rT$_3$ (24.1 ng/dL) was used as a cut-off for results above the normal range.

**Data availability statement:** The data used for the findings is available from the BioStudies repository at https://www.ebi.ac.uk/biostudies/studies/S-BSST1973 (DOI: 10.6019/S-BSST1973).

**Funding:** This research was funded by the National Institute of Drug Abuse in the form of NIDA grants, R25DA050723 and R25DA057713 to TCF, and MIMHD S21 MD000103 to Charles R. Drew University. The funders had no role in study design, data collection, and analysis, decision to publish, or preparation of the manuscript.

**Competing interests:** The authors have declared that no competing interests exist.

## Results

The number of patients with $rT_3$ levels above normal range varied significantly with the type of thyroid hormone replacement prescribed. The highest rate of an elevated $rT_3$ was 20.9% (29/139) in patients taking $T_4$ alone. Nine% (31/345) of patients not taking thyroid hormone replacement had elevated $rT_3$. Patients on all types of L-$T_4$ treatment had higher $rT_3$ levels than those not on L-$T_4$ treatment ($p < 0.00001$) and they also had a higher percentage of $rT_3$ levels above the cutoff of 24.1 ng/dL ($p < 0.00001$). Linear regression analysis showed $rT_3$ levels correlated with free $T_4$ and free $T_3$ levels and inversely with log TSH levels.

## Conclusions

This study found elevated $rT_3$ levels in patients with symptoms of fatigue on various thyroid hormone replacements with the highest levels of $rT_3$ in those taking L-$T_4$ replacement alone and the lowest levels of $rT_3$ in those on preparations that contained L-$T_3$ alone.

## 1. Introduction

Under physiological conditions, $T_4$ (thyroxine) is primarily monodeiodoniated to $T_3$ (3,3´,5-triiodo-L-thyronine) or reverse $T_3$ ($rT_3$; 3,3´,5´-triiodo-L-thyronine), depending on the energy or $T_3$ needs of the body [1]. $T_3$ is considered the active hormone because its affinity for thyroid hormone receptors is 15 times higher than $T_4$, while $rT_3$ inhibits the effects of $T_3$ and $T_4$ without binding to nuclear thyroid hormone receptors [2–5]. The clinical significance of $rT_3$ has been debated since the 1970s, when the newly adopted "thyroid function test" allowed endocrinologists to realize that severe illness causes a reduction in $T_3$ and an increase in $rT_3$ [6]. Endocrinologists termed this "euthyroid sick syndrome" and noted that it was common in many types of chronic diseases, especially in patients hospitalized in intensive care units. Whether this is an adaptive response (to conserve energy) or a pathological response (where illness leads to reduced $T_3$ production below that what is needed, thus warranting thyroid hormone treatment) is still the subject of intense debate. Overall, it is viewed that these patients should not be treated with thyroid medication [7–9].

As for the clinical significance of $rT_3$ in thyroidal illness, the data is sparse. In 1977, Burman et al. developed one of the first $rT_3$ assays and demonstrated that its levels varied significantly based on a person's thyroid status (normal, hyperthyroid, and hypothyroid), but also based on the dosage of levothyroxine that patients received [10]. In particular, they showed that patients that were hypothyroid and receiving 0.05 mg per day of levothyroxine (suboptimal dosage) had below normal $rT_3$ levels, while patients receiving 0.4 mg per day (supraoptimal dosage) had above normal $rT_3$ levels, suggesting that knowledge of $rT_3$ levels could be useful in the management of hypothyroidism. However, commercial $rT_3$ assays were not available and the measurement of $rT_3$ was confined to research settings. In 1995, Burmeister et al.

published her evaluation of 246 patients whose $rT_3$ levels were measured while being treated in a university teaching hospital. She showed $rT_3$ levels varied tremendously and judged its measurement to be unreliable in distinguishing between hypothyroid sick patients and the euthyroid sick patients [11]. Even though this work looked at the variance of $rT_3$ levels in nonthyroidal illness, many in the field felt that this unreliability extended to thyroid illness, and it is currently viewed that the measurement of $rT_3$ is of little clinical use [12,13].

The authors believe this conclusion should be reevaluated for several reasons. Firstly, $rT_3$ measurements have recently become more accurate with the wide-spread use of mass-spectrometry in commercial laboratories [2] and are available at both Quest Diagnostics and Labcorp. Since $rT_3$ inhibits the action of the biological hormone $T_3$ at the $T_3$ receptor, knowledge of $rT_3$ levels is required to completely understand the effects of thyroid hormone administration [5,14]. Importantly, thyroid medication is the second most prescribed drug in the U.S. [15] and yet as many as 40–50% of patients with these medications do not take them as prescribed [16]. Furthermore, many patients turn to alternative doctors, including functional medicine doctors, for the management of thyroid illness. These doctors often measure $rT_3$ and use it to guide patient treatment. These providers have argued that high $rT_3$ is detrimental and can block $T_3$ from binding to the thyroid hormone receptor. With little peer-reviewed publications [17], these functional medicine doctors rely heavily on $rT_3$ levels to treat patients that may have no other laboratory findings of hypothyroidism and often prescribe them L-$T_3$-only preparations to try to lower the $rT_3$. Studies looking at $rT_3$ with valid assays are needed to determine the role of this hormone. Functional medicine doctors have proposed risk factors for elevated $rT_3$ levels including stress, depression, pain, inflammation, dieting and iron deficiency [17].

In this paper, we retrospectively analyzed initial $rT_3$ measurements from 976 consecutive patients seen by TCF from 2010 to 2021 in a private Endocrinology practice. Six hundred thirty-one patients were on varying types of thyroid hormone replacement and 345 patients were not on any thyroid hormone replacement.

## 2. Methods

### 2.1 Study population and methodology

The $rT_3$ measurements were retrospectively analyzed from initial blood tests along with other thyroid function tests from 976 consecutive patients seen by TCF from 2010 to 2021 in a private Endocrinology practice. Three hundred forty-five patients were not on thyroid hormones, 226 were on desiccated thyroid extract (DTE) (Armour thyroid, NP thyroid, Naturethroid and WP thyroid were the most common brands) and not synthetic thyroid hormones, 15 were on desiccated thyroid and L-$T_3$, 138 were on desiccated thyroid and L-$T_4$, 7 were on desiccated thyroid, L-$T_3$ and L-$T_4$, 23 were on L-$T_3$ alone, 139 were on L-$T_4$ alone, and 83 were on L-$T_3$ and L-$T_4$.

All patients had fatigue as one of their main symptoms and none had a severe chronic disease in which they would be considered "sick euthyroid." All patients had $rT_3$, free $T_3$, free $T_4$, anti-thyroid peroxidase (anti-TPO), and TSH measured, usually in the morning after their visit at either Quest Diagnostics or Labcorp. $rT_3$ at both laboratories were done by liquid chromatography/tandem mass spectrometry (LC/MS-MS). JW and ME. performed chart review and were not able to identify free $T_3$, free $T_4$, anti-TPO, and TSH in several patients during their chart review. The normal range for $rT_3$ at Labcorp was 9.1 to 24.1 ng/dL and Quest Diagnostics was 8.0 to 25.0 ng/dL. The cut-off for results above the range used the value of 24.1 ng/dL and these values were determined in euthyroid patients. TSH, free $T_3$ and free $T_4$ were measured by electrochemiluminescence immunoassay (ECLIA) with a range of 0.45- 4.5 miU/mL, 2.0 to 4.4 pg/mL and 0.82 − 1.77 ng/dL, respectively at Labcorp and 0.45- 4.5 miU/mL, 2.3-4.2 pg/mL and 0.8-1.8 ng/dL at Quest. Anti-TPO was done by chemiluminescense at Esoterix Laboratories (subsidiary of Labcorp) and Quest with a range of < 9.0 IU/mL at both laboratories.

### 2.2 Statistical analyses

Sub-analyses of the pairwise comparisons of the thyroid treatment groups, and significance of anti-TPO status within these groups were performed using Dunn's test. The Wilcoxon rank-sum test was used to compare $rT_3$ levels between all

groups on L-T$_4$ treatment and all groups not on L-T$_4$ treatment. The chi-square test for homogeneity was used to compare the % of patients with rT$_3$ above range between all groups on L-T$_4$ treatment and all groups not on L-T$_4$ treatment. Pearson correlations between rT3, free T3, free T4 and log TSH (to compensate for severe non-normality) levels were calculated and the significance of these (from zero) were determined by the appropriate t-test. The patients with circulating anti-TPO antibodies who had elevated rT$_3$ levels were compared to patients without these antibodies or who did not have them tested were compared by the Fisher's exact test.

## 2.3 IRB approval

The Charles R. Drew University of Medicine and Science (CDU) Institutional Review Board (IRB) approved this retrospective study under Exemption Category # 4 (45CFR46.104, category 4iii). The most recent approval date was January 3, 2024. The data was accessed for research purposes on January 18, 2022, and accessed again on January 1, 2024. The authors had access to information that could identify individual participants during or after data collection, however the CDU IRB approved the use of PHI as involving no more than minimal risk and did not require a waiver of consent.

## 2.4 Patient and public involvement

Patients and the public were not involved in the design or the interpretation of the study although patients in this study have informed the investigators about the importance of measuring rT$_3$.

## 3. Results

Table 1 shows the Mean, SD, and N of rT$_3$, free T$_3$, free T$_4$ and TSH in the study population. 810 patients were female and 166 were male. Overall, 107 (11.0) patients had an elevated rT$_3$ value. The proportion of patients with above normal rT$_3$ values was found to be significantly affected by treatment (Fig 1) with the highest rate of elevated rT$_3$ in 20.9% (29/139) of patients taking T$_4$ alone. Nine percent (31/345) of patients not taking thyroid hormone replacement had elevated rT$_3$ values. In contrast, only 3.5% (8/226) of patients taking desiccated thyroid hormone had above normal rT$_3$ values, compared to 12% (10/83) of patients taking a T$_3$-T$_4$ combination and 17.4% (24/138) of patients taking desiccated thyroid-T$_4$ combination. The proportion of patients taking desiccated thyroid hormone replacement with above normal rT$_3$ levels was found to be significantly less than all other groups except for patients taking a desiccated-T$_3$ combination (Fig 1). Table 2 shows the P-values of pairwise comparisons of the groups using Dunn's test.

Table 3 shows that patients on all types of L-T$_4$ treatment had higher mean rT$_3$ levels than those not on L-T$_4$ treatment (p < 0.0001) and a higher percentage of rT$_3$ levels above the cutoff of 24.1ng/dL (p < 0.0001). Linear regression analysis (Table 4) showed rT$_3$ levels strongly correlated with free T$_4$ and free T$_3$ levels and inversely with log TSH levels.

The presence of anti-TPO antibodies was assessed in 712 of these patients with 212 patients having anti-TPO antibodies above the range and 500 patients not having elevated levels. For the patients not on thyroid hormone replacement, 41 of 345 patients had anti-TPO antibodies and 304 patients did not have antibodies or were not assessed. For the patients on thyroid hormone replacement, 170 of 631 patients had anti-TPO antibodies and 461 patients did not have antibodies or were not assessed. For most thyroid treatment regimens, the proportion of patients with above normal rT$_3$ levels did not vary significantly with TPO antibody status, except for patients taking desiccated thyroid hormone replacement. For this

**Table 1. Mean, SD, N of rT$_3$ (ng/dL), Free T$_3$ (pg/mL), Free T$_4$ (ng/dL) and TSH (µIU/mL).**

|  | rT$_3$ | FreeT$_3$ | FreeT$_4$ | TSH |
|---|---|---|---|---|
| Mean | 16.4 | 3.2 | 1.2 | 2.3 |
| SD | 6.6 | 1.0 | 0.3 | 4.3 |
| N | 976 | 452 | 455 | 513 |

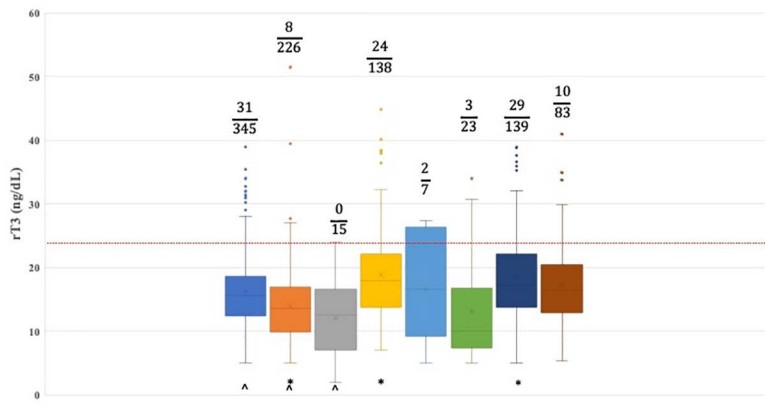

**Fig 1. Box and whisker plot of patient rT$_3$ based on the type of thyroid hormone replacement.** None = Not taking any thyroid hormone replacement; Des = Desiccated thyroid hormone replacement; Des/T$_3$ = Desiccated thyroid-T$_3$ combination; Des/T$_4$ = Desiccated thyroid-T$_4$ combination; Des/T$_3$/T$_4$ = Desiccated thyroid-T$_3$-T$_4$ combination; T$_3$/T$_4$ = T$_3$-T$_4$ combination. * P < 0.05 compared to none. ^ P < 0.05 compared to T$_4$.

**Table 2. P-values of pairwise comparisons of the groups using Dunn's test.**

|  | None | Des | Des/T$_3$ | Des/T$_4$ | Des/T$_3$/T$_4$ | T$_3$ | T$_4$ | T3/T4 |
|---|---|---|---|---|---|---|---|---|
| **None** | – | 0.01 | 0.22 | **0.01** | 0.08 | 0.52 | **$3x10^{-4}$** | 0.39 |
| **Des** | **0.01** | – | 0.46 | **$6x10^{-6}$** | **$1x10^{-3}$** | **0.03** | **$1x10^{-7}$** | **$5x10^{-3}$** |
| **Des/T$_3$** | 0.22 | 0.46 | – | 0.08 | **0.03** | 0.14 | **0.05** | 0.16 |
| **Des/T$_4$** | **0.01** | **$6x10^{-6}$** | 0.08 | – | 0.45 | 0.61 | 0.46 | 0.29 |
| **Des/T$_3$/T$_4$** | 0.08 | **$1x10^{-3}$** | **0.03** | 0.45 | – | 0.33 | 0.63 | 0.22 |
| **T$_3$** | 0.52 | **0.03** | 0.14 | 0.61 | 0.33 | – | 0.38 | 0.90 |
| **T$_4$** | **$3x10^{-4}$** | **$1x10^{-7}$** | **0.05** | 0.46 | 0.63 | 0.38 | – | 0.09 |
| **T3/T4** | 0.39 | **$5x10^{-3}$** | 0.16 | 0.29 | 0.22 | 0.90 | 0.09 | – |

None = Not taking any thyroid hormone replacement; Des = Desiccated thyroid hormone replacement; Des/T$_3$ = Desiccated thyroid-T$_3$ combination; Des/T$_4$ = Desiccated thyroid-T$_4$ combination; Des/T$_3$/T$_4$ = Desiccated thyroid-T$_3$-T$_4$ combination; T$_3$/T$_4$ = T$_3$-T$_4$ combination.

**Table 3. Effect of L-T$_4$ treatment on rT$_3$ levels.**

|  | All groups on L-T$_4$ treatment | All groups not on L-T$_4$ treatment | p-value |
|---|---|---|---|
| rT$_3$ (Mean ± SD) | 18.4 ± 7.1 | 15.2 ± 6.0 | <0.00001* |
| % with rT$_3$ > 24.1 | 65/367 (17.7%) | 42/609 (6.9%) | p = 1.63 x 10$^{-7}$^ |

*Wilcoxon rank-sum test, ^chi-square test for homogeneity

**Table 4. Correlations between rT$_3$ levels and other hormones.**

|  | FreeT$_3$ | FreeT$_4$ | Log TSH |
|---|---|---|---|
| Pearson correlation | 0.184 | 0.624 | −0.298 |
| p | 0.0001 | <0.0001 | <0.0001 |

group, 0% (0/67) of the patients with circulating anti-TPO antibodies had elevated rT$_3$ levels, while 7% (6/88) of patients without these antibodies or who did not have them tested had elevated rT$_3$ levels (p = 0.037). For those taking thyroid hormone preparations besides DTE, 20 of 103 had anti-TPO antibodies and 49 of 270 did not have antibodies or were not assessed (p = NS).

## 4. Discussion

The initial rT$_3$ measurements were retrospectively analyzed from 976 consecutive patients before management by TCF. Patients with hypothyroidism generally sought out TCF due to dissatisfaction with their current management, including persistent fatigue despite being on what their previous provider considered adequate treatment. We did not track what type of providers treated these patients but posit that it included primary care providers, other Endocrinologists, functional medicine doctors, mid-level health care providers (i.e., physician assistants, nurse practitioners, etc.), and holistic doctors. The high prevalence of patients treated with thyroid preparations other than L-T$_4$ reflects the heterogeneity of providers as well as the dissatisfaction with conventional treatment among this group of patients. We found that the proportion of patients with above normal rT$_3$ values varied significantly depending on the type of thyroid medication they were taking. This proportion was higher in patients taking preparations containing L-T$_4$ but was lower in patients taking desiccated thyroid or L-T$_3$ preparations. Groups that took preparations containing desiccated thyroid and/or L-T$_3$ with L-T$_4$ had a larger proportion of patients with above normal rT$_3$ values than groups that took the same preparations without L-T$_4$. The group taking L-T$_4$ alone had the highest percentage of elevated rT$_3$ values at 20.9%. Our results are consistent with previous findings that in short-term settings, L-T$_4$ can raise rT$_3$ levels [18] and L-T$_3$ can lower rT$_3$ levels [19], although these effects need to be verified prospectively in larger, newer studies.

Although the majority of patients do have a satisfactory response on L-T$_4$ therapy, up to 15% of properly treated hypothyroid patients fail to achieve a sense of well-being on levothyroxine and continue to have hypothyroid symptoms despite normalized thyrotropin levels. The causes of patients' lack of well-being have been discussed including by the American Thyroid Association Task Force on Thyroid Hormone Replacement [20] and include decreased serum T$_3$/T$_4$ ratio and alterations in the inherited *DIO2* polymorphism [20,21]. Combination therapy with L-T$_4$ plus L-T$_3$ has been found to be helpful in some, but not all studies [22]. One review proposed that some patients likely have a compounding condition that increases the likelihood of developing symptoms [21]. Could this be elevated rT$_3$ levels?

It is estimated that about 10–29% of patients with hypothyroidism use DTE as their primary thyroid hormone replacement medication in the US [23–25] despite concerns about the potential risk of thyrotoxicosis associated with DTE use [26,27]. Toleza et al. surveyed the content of online posts from three popular hypothyroidism forums from patients currently taking DTE and found the most frequently described benefits associated with DTE use were an improvement in symptoms (56%) and a change in overall well-being (34%) [25].

A 2013 crossover study by Hoang and colleagues compared levothyroxine to a DTE preparation (Armour Thyroid) [28]. They used 70 patients that were enrolled in a military healthcare system, were on a stable dose of levothyroxine, and had a normal TSH before the study started. During the study, patients lost an average of three pounds during once-a-day Armour Thyroid therapy, and at the conclusion of the study, they found that 49% (34/70) preferred Armour Thyroid, 19% (13/70) preferred levothyroxine, and 33% (23/70) had no preference. Importantly, patients had thyroid function tests performed at the beginning of the study and they compared the patients' initial rT$_3$ levels to ultimate preference for thyroid medication. The baseline (on L-T$_4$) rT$_3$ level was 32.3 ± 12.9 ng/dL that stayed elevated at 31.4 ± 12.1 ng/dL after receiving L-T$_4$ but decreased to 21.1 ± 10.9 ng/dL (p < 0.001) following DTE treatment. rT$_3$ was measured by RIA at Radim in Pomezia, Italy with the range not given. This prospective cross-over study supports that DTE lowers rT$_3$ levels and is preferred by the majority of patients, although causality between rT$_3$ levels and patient preference for DTE was not established and needs to be examined in larger studies.

This study was confirmed by Shakir and colleagues [29] who randomized patients to L-T$_4$, L-T$_4$+L-T$_3$, or DTE for 22 weeks. They found quality of life outcomes were similar among hypothyroid patients taking DTE vs L-T$_4$+L-T$_3$ or

L-T$_4$. However, those patients that were most symptomatic on L-T$_4$ preferred and responded positively to therapy with L-T$_4$+L-T$_3$ or DTE. In support of our data, rT$_3$ levels were highest in L-T$_4$ treatment, lowest in DTE-treated patients and in the middle for those on L-T$_4$+L-T$_3$ ($p < 0.001$). They did not find any difference between the rT3 levels in autoimmune and non-autoimmune patients and did not analyze if the rT3 levels correlated with symptoms.

The strengths of our study include a large number of subjects taking different types of thyroid preparations, similar to what many patients are taking in real world settings. Another strength is the measurement of rT$_3$ levels with an accurate mass-spectrometry methodology. The limitations of the study include a potential bias of subjects more inclined to taking L-T$_4$+L-T$_3$ and DTE preparations than would be seen in a typical Endocrinology clinic. However, our findings of the highest rT$_3$ levels in those on L-T$_4$ alone would favor higher rT$_3$ levels in a typical Endocrinology clinic in which most patients are on L-T$_4$ alone. Other limitations include the retrospective nature of the study, lack of objective measurements of fatigue and quality of life (QOL), and the lack of a causal relationship between rT$_3$ levels and fatigue and QOL symptoms, a limitation that should be addressed in future studies.

## 5. Conclusions and future studies

In conclusion, our study found elevated rT$_3$ levels in patients with symptoms of fatigue on various thyroid hormone replacements with the highest levels seen in patients on L-T$_4$ replacement alone and the lowest levels seen in those on preparations that contain L-T$_3$, including DTE. It would be premature to conclude that elevated rT$_3$ levels are the cause of the symptoms in approximately 15% of patients on L-T$_4$ replacement, and further studies are needed to understand the relationship better. Nine percent of patients not taking thyroid hormone replacement in our study had elevated rT$_3$ values.

Further studies are needed to understand the implications of elevated rT$_3$ values in patients both on and off thyroid hormone replacement and whether its measurement will be useful in clinical practice to guide thyroid hormone replacement. Randomized control studies are needed to determine if treatment with DTE or L-T$_3$ in patients with elevated rT$_3$ values will both lower elevated rT$_3$ values and improve measurements of fatigue and QOL. Additional further studies are needed to determine what factors raise rT$_3$ values and if correcting them improves hypothyroid symptoms. Overall, our study will open new avenues of thyroid disease research that could lead to improvement in clinical outcomes in patients with hypothyroidism.

## Acknowledgments

We thank Robin Faria of the CTSI Grants Submission Unit of the UCLA Clinical and Translational Science Institute (grant number UL1TR001881) for editing this manuscript.

Raw data used for the findings is available in the following public repository:
https://www.ebi.ac.uk/biostudies/studies/S-BSST1973

## Author contributions

**Conceptualization:** Theodore Friedman.

**Data curation:** Theodore Friedman, Julian B. Wilson.

**Formal analysis:** Theodore Friedman, Julian B. Wilson, Thanh D. Hoang, Martin L. Lee, Ma'ayan Epstein.

**Investigation:** Theodore Friedman.

**Methodology:** Theodore Friedman, Martin L. Lee.

**Supervision:** Theodore Friedman.

**Writing – original draft:** Theodore Friedman, Julian B. Wilson.

**Writing – review & editing:** Theodore Friedman, Julian B. Wilson, Thanh D. Hoang, Martin L. Lee, Ma'ayan Epstein.

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
