## [Decision Letter · Decision Letter 0]

24 Mar 2025

Dear Dr. Friedman,

Thank you for submitting your manuscript to PLOS ONE. After careful consideration, we feel that it has merit but does not fully meet PLOS ONE’s publication criteria as it currently stands. Therefore, we invite you to submit a revised version of the manuscript that addresses the points raised during the review process.

We look forward to receiving your revised manuscript.

Kind regards,

Apeksha Niraula, M.D.

Academic Editor

PLOS ONE

**Journal Requirements:**

1. When submitting your revision, we need you to address these additional requirements. Please ensure that your manuscript meets PLOS ONE's style requirements, including those for file naming. The PLOS ONE style templates can be found at https://journals.plos.org/plosone/s/file?id=wjVg/PLOSOne_formatting_sample_main_body.pdf and https://journals.plos.org/plosone/s/file?id=ba62/PLOSOne_formatting_sample_title_authors_affiliations.pdf 2. In the online submission form, you indicated that The data underlying the results presented in the study are available from the corresponding author. All PLOS journals now require all data underlying the findings described in their manuscript to be freely available to other researchers, either a. In a public repository, b. Within the manuscript itself, or c. Uploaded as supplementary information.This policy applies to all data except where public deposition would breach compliance with the protocol approved by your research ethics board. If your data cannot be made publicly available for ethical or legal reasons (e.g., public availability would compromise patient privacy), please explain your reasons on resubmission and your exemption request will be escalated for approval.

Reviewers' comments:

Reviewer's Responses to Questions

**Comments to the Author**

1. Is the manuscript technically sound, and do the data support the conclusions?

Reviewer #1: Partly

2. Has the statistical analysis been performed appropriately and rigorously?

Reviewer #1: I Don't Know

3. Have the authors made all data underlying the findings in their manuscript fully available?

Reviewer #1: Yes

4. Is the manuscript presented in an intelligible fashion and written in standard English?

Reviewer #1: No

**Reviewer #1: ** 1. Do not start sentence with digit and write single digit in a words.

2.Revise the abstract: Background: Check these sentence "Decades ago, endocrinologists realized that in severe illnesses" and

"Without peer-reviewed publications, these functional medicine doctors rely heavily on rT3 levels to treat patients that may have no other laboratory findings".

3. Background ended with objective of the study

4. Kindly elaborate the methods (In abstract): sample collection, which sample was used, which methods were used to measure fT3, fT4 and TSH.

5. Results (in abstract): Revise and mention the major findings only

6. Rewrite your conclusion (in abstract) based on the objective and findings of the study and remove the recommendation/suggestions from conclusion section.

7. The first paragraph of the result section includes the subgroup of the study population. It is better to mention it in the materials and methods.

8. Kindly mentioned the name statistical tests applied in the result section to test the significance.

9. Kindly revise the result section with appropriate table and figure. Are same data presented in table 2 and figure 1? Kindly explained it.

9. Write the conclusion in a separate section.

**Do you want your identity to be public for this peer review?** For information about this choice, including consent withdrawal, please see our Privacy Policy

Reviewer #1: No

---

## [Author Response · Author response to Decision Letter 1]

18 Apr 2025

Department of Internal Medicine

April 3, 2025

Apeksha Niraula, M.D.

Academic Editor

PLOS ONE

RE: PONE-D-24-59271

Reverse T3 in patients with hypothyroidism on different thyroid hormone replacement

Dear Dr. Niraula. We thank you and the reviewer for his/her helpful comments on our manuscript entitled “Reverse T3 in patients with hypothyroidism on different thyroid hormone replacement”. Our point-by-point response to the reviewer’s comments are below with our response in italics.

We have included the following items in our revised submission:

1. A rebuttal letter that responds to each point raised by the academic editor and reviewer(s) labeled 'Response to Reviewers'.

2. A marked-up copy of our manuscript that highlights changes made to the original version labeled 'Revised Manuscript with Track Changes'.

3) An unmarked version of our revised paper without tracked changes labeled 'Manuscript'.

Comments to the Author

1. Is the manuscript technically sound, and do the data support the conclusions?

The manuscript must describe a technically sound piece of scientific research with data that supports the conclusions. Experiments must have been conducted rigorously, with appropriate controls, replication, and sample sizes. The conclusions must be drawn appropriately based on the data presented. Reviewer #1: Partly

We carefully reviewed the manuscript and can state that our manuscript a technically sound piece of scientific research with data that supports the conclusions. Our experiments have been conducted rigorously, with appropriate controls, replication, and sample sizes.

2. Has the statistical analysis been performed appropriately and rigorously? Reviewer #1: I Don't Know

Our statistician, Martin l. Lee, PhD has reviewed the statistical portions of the manuscript and concludes that the statistical analysis been performed appropriately and rigorously.

3. Have the authors made all data underlying the findings in their manuscript fully available? Reviewer #1: Yes

Thank you.

4. Is the manuscript presented in an intelligible fashion and written in standard English?

PLOS ONE does not copyedit accepted manuscripts, so the language in submitted articles must be clear, correct, and unambiguous. Any typographical or grammatical errors should be corrected at revision, so please note any specific errors here. Reviewer #1: No

We had a professional grant writer review the manuscript and have included her corrections in the new version.

5. Review Comments to the Author

Reviewer #1:

1. Do not start sentence with digit and write single digit in a words. We have changed that.

2. Revise the abstract: Background: Check these sentence "Decades ago, endocrinologists realized that in severe illnesses" and

"Without peer-reviewed publications, these functional medicine doctors rely heavily on rT3 levels to treat patients that may have no other laboratory findings". We have changed these sentences.

3. Background ended with objective of the study. We checked and the background in the abstract did not include the objective of the study

4. Kindly elaborate the methods (In abstract): sample collection, which sample was used, which methods were used to measure fT3, fT4 and TSH. This information has been added to the Methods section of the Abstract.

5. Results (in abstract): Revise and mention the major findings only. We have revised the Results section of the Abstract to only include major findings.

6. Rewrite your conclusion (in abstract) based on the objective and findings of the study and

remove the recommendation/suggestions from conclusion section. We have removed the last sentence of the Conclusions section.

7. The first paragraph of the result section includes the subgroup of the study population. It is better to mention it in the materials and methods. We have moved the subgroup of the study population to the Methods section.

8. Kindly mentioned the name statistical tests applied in the result section to test the significance. The statistical tests are described in the 2.2 Statistical Analyses. We have moved the mention of the Fisher’s exact test to the Methods section.

9. Kindly revise the result section with appropriate table and figure. We have removed Table 2 and renumbered the remaining table numbers. Are same data presented in table 2 and figure 1? We agree that Table 2 and Figure 1 are duplications and have removed Table 2.

10. Write the conclusion in a separate section. We have added a Conclusions and Future Studies Section.

We look forward to hearing from you after the re-review process.

Sincerely yours,

Theodore C. Friedman, M.D., Ph.D.

Chairman, Department of Internal Medicine

Chief, Division of Endocrinology, Metabolism and Molecular Medicine

Professor of Medicine-Charles R. Drew University of Medicine & Science

1731 E. 120th St. Los Angeles, CA 90059

Professor of Medicine UCLA

theodorefriedman@cdrewu.edu

phone (323) 539-8020

fax (310) 564-2786

---

## [Editor Report · Decision Letter 1]

7 May 2025

Reverse T3 in patients with hypothyroidism on different thyroid hormone replacement

PONE-D-24-59271R1

Dear Dr. Friedman,

We’re pleased to inform you that your manuscript has been judged scientifically suitable for publication and will be formally accepted for publication once it meets all outstanding technical requirements.

Kind regards,

Apeksha Niraula, M.D.

Academic Editor

PLOS ONE
---

## [Editor Report · Acceptance letter]

PONE-D-24-59271R1

PLOS ONE

Dear Dr. Friedman,

I'm pleased to inform you that your manuscript has been deemed suitable for publication in PLOS ONE. Congratulations! Your manuscript is now being handed over to our production team.

Kind regards,

on behalf of

Dr. Apeksha Niraula

Academic Editor

PLOS ONE